# Study on the Static-Bending Properties of Surface-Reinforced Wood with Asymmetric Fibers

Xu Guo [1], Fengwei Zhou [1], Shuduan Deng [2],* and Chunlei Dong [1],*

1   Collage of Materials and Chemical Engineering, Southwest Forestry University, Kunming 650224, China; gx0120@swfu.edu.cn (X.G.); fengweizhou@swfu.edu.cn (F.Z.)
2   New Wood-Based Composites Innovation Team, Southwest Forestry University, Kunming 650224, China
*   Correspondence: dengshuduan@163.com (S.D.); dchunlei@swfu.edu.cn (C.D.)

**Abstract:** In order to investigate the mechanism of the effect of asymmetric reinforcement on the static-bending properties of wood, this paper tests and analyzes the static-bending properties of SPF wood and seven different types of asymmetric fiber surface-reinforced wood (AFRWC) formed by SPF wood as the substrate and bamboo and carbon fibers as the reinforcement materials. The results of the study found that (1) the moduli of rupture of the seven types of AFRWC were increased to varying degrees, but the static-bending moduli of elasticity increased or decreased; (2) the asymmetric reinforcement changed the cross-section strain distribution and damage type of the wood in static bending; (3) the results of the cross-section strain-field tests and the ABAQUS finite element simulation showed that the asymmetric reinforcement method of bonding the bamboo material and the two layers of CFRP in the compression and tensile zones, respectively, can greatly enhance the static-bending performance of the wood. The error between the simulated and measured values of specimens MOR and MOE is only −0.7% and −7.3%, respectively. This type of asymmetric reinforcement makes it possible to obtain a more reasonable cross-section stress distribution.

**Keywords:** asymmetric fiber reinforcement; CFRP; bamboo–wood composite; static-bending performance

## 1. Introduction

Compared with common inorganic materials, such as carbon fiber and glass fiber, and metallic materials, such as steel, and biomass materials such as bamboo, the strength and modulus of elasticity of wood are much lower, and, thus, low-grade commercial-grade wood is rarely used directly as load-bearing or large-span wood members [1,2]. The use of materials such as carbon fiber, glass fiber, and bamboo composites with wood can increase the strength and modulus of elasticity of wood, thus greatly expanding its application in the field of large-span and high-strength load-bearing building materials [3–6]. At present, by the definition of the engineering community on laminated composite materials, the poor material of the fast-growing timber side (generally for the tensile side), adhesive glass fiber, carbon fiber, bamboo fiber, or other high-strength fiber materials, and the composite material obtained is called asymmetric fiber-reinforced wood composite (AFRWC). Compared with ordinary wood, AFRWC makes up for the natural material defects of ordinary wood, which has generally weak tensile properties, and greatly improves the mechanical properties of wood because the reinforcing fibers provide local bridging to defective parts of the wood (e.g., cracks within drying, knots, etc.), limiting the local fracture of the wood, and preventing the cracks from expanding [7]. Nadir et al. [8] used carbon-fiber-reinforced polymer (CFRP) sheets bonded to the tensile side of laminated timber beams for composite reinforcement, which enhanced the static-bending strength and modulus of the log beams by 51% and 64%, respectively. The more layers of sheets that were used, the more pronounced the enhancement effect was. Chun Qing et al. [9] used carbon-fiber-reinforced polymer to enhance the bending capacity of pine

beams to improve the beam's bending capacity by 12.9%–34.5% and the form of damage for the bottom surface of the timber fracture. But, the carbon fiber has not yet been destroyed, the whole beam still has a certain load-bearing capacity with nonbrittle damage, and the safety of the beam relative to the log beams has been greatly improved.

When compared with inorganic and metallic materials, bamboo has an average of 2–3 times higher tensile and compressive strengths and moduli than ordinary wood [10]. It is a natural high-strength fiber material with a very short maturing cycle (generally only 5 years), has a large production volume in China, and has an organic gluing interface of a similar nature when composited with wood for easier gluing [11]. So, it has gained significant attention from both industry and academia as an ideal reinforcing fiber material. In the past, scholars have used bamboo-laminated panels [12] and bamboo scrimber [4,13] to composite with wood to form bamboo–wood composite materials. For example, Wei Y et al. [14] used bamboo scrimber and CFRP attached to the tensile region at the bottom of wood beams and investigated its effect on the mechanical properties of the whole beam. They found that the static-bending strength and modulus of the wood beams were enhanced by 50%–90% and 70%–130%, respectively, with the increase in the number of CFRP layers, but the increase in the thickness of the bamboo scrimber material had no significant effect on the static flexural strength and modulus of the wood beams. Si Chen et al. [15] used Douglas fir as the web and a combination of bamboo scrimber and fir as the flanges to test the bending performance of bamboo–wood I beams, and the static-bending strength and modulus of elasticity of bamboo I beams increased by 44.8% and 23.4% on average compared with that of control wood beams. Leng Yubing [16] found that the static-bending strength of bamboo-glued beams and glued bamboo–wood beams were 28% and 38% higher than that of Douglas fir-glued wood beams, but the static-bending modulus of elasticity of the two was lower than that of glued wood beams.

When wood is subjected to static-bending loads, its upper surface is mainly under compressive stress, while its lower surface is mainly exposed to tensile stress. Therefore, gluing materials with a greater strength and modulus than wood to the upper and lower surfaces of the wood can improve its static-bending properties; this is how the mechanical properties of wood can be improved by laminating its surface with other high-strength materials. The tensile and compressive properties of wood differ from those of other materials [17]. Therefore, simply reinforcing the wood by gluing a material on both sides of the top and bottom surfaces of the beams, or unilaterally on the bottom surface [18,19], may not fully utilize its composite reinforcement effect. This paper aims to investigate and compare the effects of two different reinforcing materials, carbon fiber and bamboo, on the static-bending properties and damage forms of wood. Additionally, the study will focus on understanding the mechanism of asymmetric reinforcement on the strain field of the AFRWC cross section. The ultimate goal is to provide a reference for the optimal design of laminated wood composites.

## 2. Materials and Methods

### 2.1. Materials and the Basic Physical and Mechanical Properties

The wood was selected from SPF (spruce, pine, fir) produced by the Weyerhaeuser Company, Edmonton, AB, Canada. The specimens were sawn into small specimens with dimensions (L × R × T) of 400 mm × 20 mm × 20 mm, and the basic physical and mechanical properties were tested by the GB/T 1927.10-2021 [20]. At the same time, 54 specimens with a similar static flexural modulus of elasticity were prepared for the test. The bamboo material was selected from Fujian Jianou (*Phyllostachys pubescens*), heat-treated at 180 °C for 30 min, and the thickness of the bamboo board was made to be 5 mm after removing the bamboo outer skin and inner skin. Finally, it was sawed into a bamboo board with the size (length × width × thickness) of 400 mm × 20 mm × 5 mm. According to GB/T15780-1995 [21], Experimental Methods of Physical and Mechanical Properties of Bamboo Materials, the physical and mechanical properties of the bamboo panels were tested. The CFRP was T300 produced by Shanghai Zhinuo Decorative Materials Co. (Shanghai, China). The adhesive is E51 bisphenol A epoxy resin and the curing agent

is produced by Jiangxi Nanchang Chenfang Adhesive Products Co. (Nanchang, China) Table 1 is a summary of the basic properties of the three test materials.

**Table 1.** Basic Properties of Materials.

| Material | Source | Density g/cm³ | Static-Bending Modulus of Elasticity /MPa | Static Strength /MPa | Tensile Strength /MPa | Tensile Modulus /MPa |
|---|---|---|---|---|---|---|
| SPF | Canada | 0.408 | 9636 | 68 | 66 | 11,336 |
| Bamboo | Jianou, Fujian | 0.532 | - | 84 | 74 | 6084 |
| CFRP | Shanghai | 1.796 | - | - | >3000 | >210,000 |

### 2.2. Specimen Production

In this experiment, CFRP and bamboo board were used to adhere to the surface of SPF to form seven different types of SPF composite lumber, and the composite method of each type of SPF composite lumber and SPF control lumber and their specification sizes and numbers are shown in Table 2. Among them, the steps for making composite materials by sticking CFRP and SPF wood are as follows.

**Table 2.** Combined design of symmetric fiber surface-reinforced composites.

| Type | W | WB | BW | BWB | WC | WC2 | BWC | BWC2 |
|---|---|---|---|---|---|---|---|---|
| Specimen size | | | | 400 mm × 20 mm × 20 mm | | | | |
| Section sketch | | | | | | | | |
| Number | 6 | 6 | 6 | 6 | 6 | 6 | 6 | 6 |

W means SPF wood; B means bamboo; C means CFRP, and C2 means adhesive two-layer CFRP.

First of all, we use 240 mesh sandpaper to sand the pasted surface of the wood, and use the homemade tensioning device to apply tension to the carbon-fiber-reinforced polymer (Figure 1); then, in Kunming, under room temperature (about 25 °C), the epoxy-resin adhesive is evenly brushed on the sanded surface of the wood and the surface of the tensioned carbon-fiber-reinforced polymer and the amount of adhesive applied is about 300 g/m². Finally, the glued surfaces of the two materials were pasted together, and the glued surfaces were subjected to a gluing pressure of about 0.5 MPa and left for 24 h to make them fully cured.

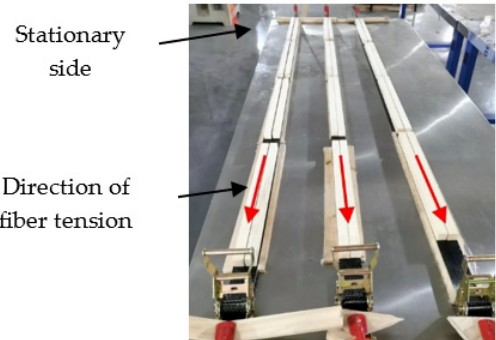

**Figure 1.** Stretching and adhesion of CFRP. (The direction of the red arrow indicates the direction of the applied tension force).

Bamboo boards and SPF wood are glued to make composite materials in the way shown in Figure 2, with a 300 g/m² sizing. The bamboo boards and wood are glued to the surface of the epoxy-resin adhesive uniformly brushed and coated using transverse fixing, and vertical pressurization of the direction of the gluing pressure is sufficient for application and is left to stand for 24 h to make it completely cured.

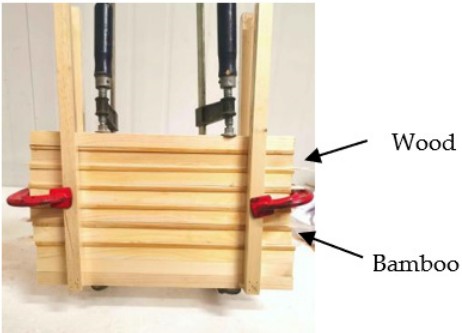

**Figure 2.** Adhesion of bamboo and wood.

*2.3. Testing of Specimens*

(1) According to the GB/T 1927.10-2021 standard four-point bending method on the above specimens for modulus of rupture (MOR) and static-bending modulus of elasticity (MOE) test. The four-point bending test allows the bending stiffness of the specimen to be evaluated without neglecting shear deformation [22]. The test span of the specimen was taken as 360 mm, and the loading speed was 2 mm/min. The MOR and MOE of the specimen were calculated according to Equations (1) and (2), respectively [20]:

$$MOR = \frac{FL}{bh^2} \tag{1}$$

$$MOE = \frac{23PL^3}{108bh^3f} \tag{2}$$

where $F$ is the maximum load (N); $L$ is the span between the two supports (mm); $b$ and $h$ are the width and height of the cross section (mm), respectively; $P$ is the load increment at the elastic stage of the load-displacement curve (N); $f$ is the cor. responding displacement at the midspan cross section (mm);

(2) During the static-bending experiments, the full-field strains within the interval between the two loading points of each specimen were collected by using a digital image correlation method (DIC, manufacturer: LINCONST TECH (Suzhou, China), Model: DSE-5M), which was used to measure the strain at the two loading points of each specimen, to analyze changes in strain in different types of specimen sections subjected to static-bending stresses. Figure 3a,b shows the loading direction, and the shaded area indicates the strain region. Figure 3c shows the directions of the wood.

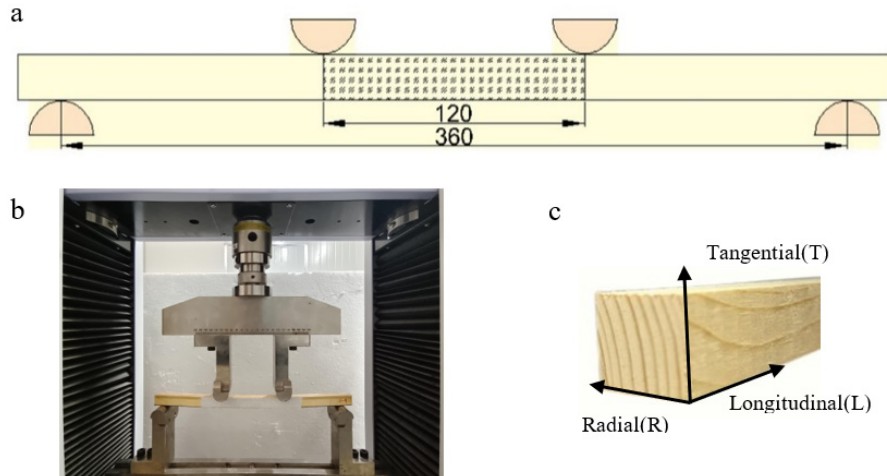

**Figure 3.** Static-bending test of specimens. (**a**) is the four-point bending schematic; (**b**) is the experimental test reality; (**c**) indicates the direction of the wood.

## 3. Results and Analysis

### 3.1. Effect of the Reinforcement Method on the Specimen Damage Form

Figures 4–6 give eight types of specimens of static-bending force until the destruction of the whole process of the "load–displacement" curve and the final form of destruction. According to the final damage form of the specimen, the damage form of the eight types of specimens can be divided into three types: brittle fracture damage, plastic fracture damage, and base-material shear damage.

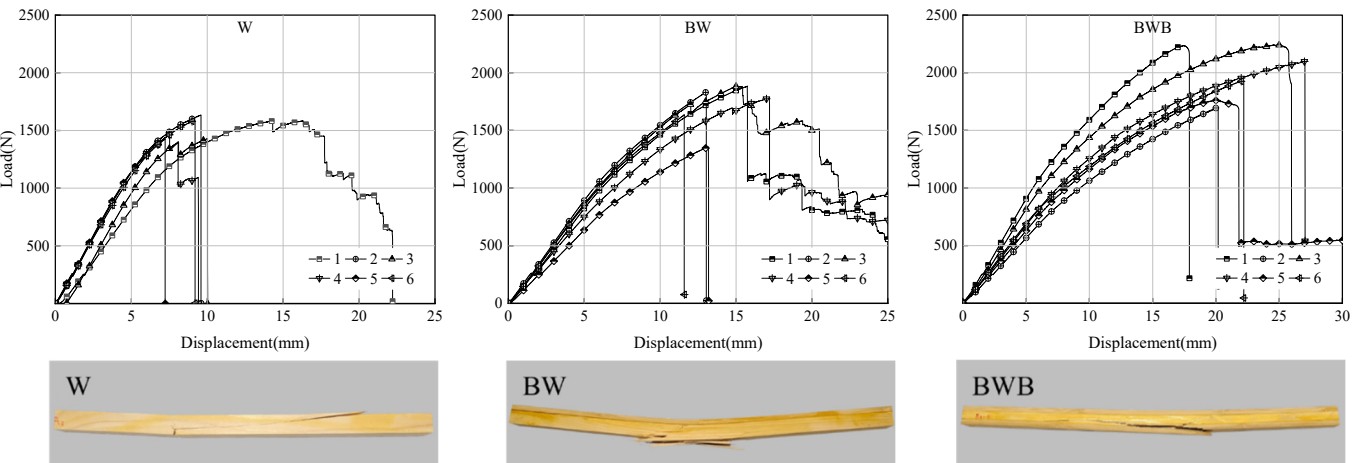

**Figure 4.** Load displacement curve and failure mode of brittle failure specimens.

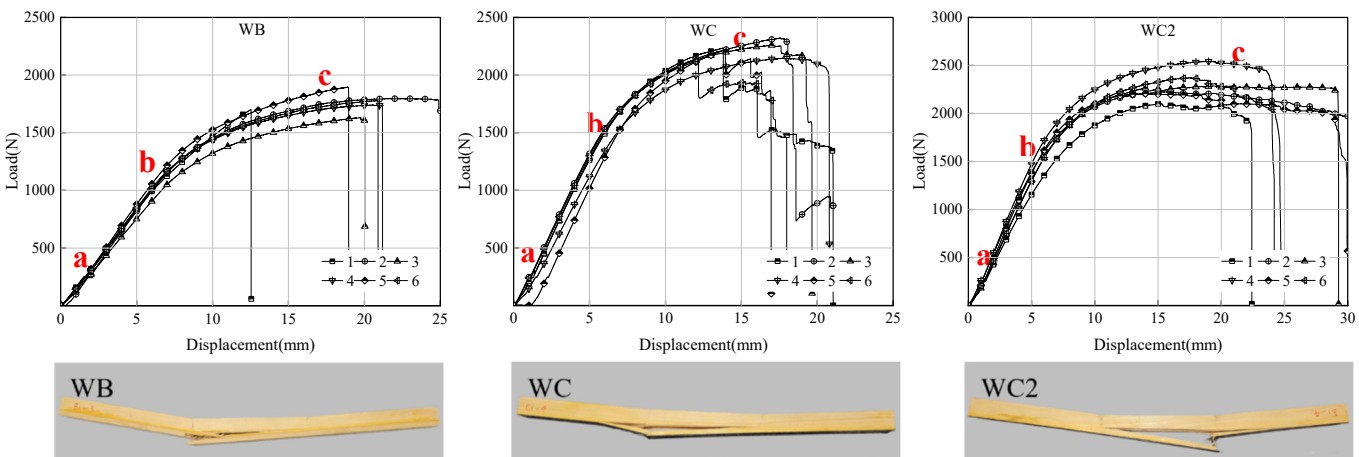

**Figure 5.** Load displacement curve and failure mode of plastic failure specimen. (During loading the specimen produces elastic deformation in sections a–b and inelastic deformation in sections b–c.)

As shown in Figure 4, the load-displacement curves of the specimens in groups W (control group), BW, and BWB exhibit a similar trend. Fiber brittle fracture damage occurred in wood and bamboo after a slight reduction in specimen stiffness. Most of the specimens in group W experienced brittle fracture damage due to tensile fracture of the bottom fibers of the SPF timber at a midspan deflection of less than 10 mm, causing rapid penetration of the cracks throughout the entire cross section. On the other hand, the BW group of specimens had an SPF compression surface (upper surface) that was reinforced with a bamboo composite. During testing, these specimens experienced sequential tensile or direct brittle fracture damage of fibers in several regions at the bottom of the SPF timber. In some specimens of the BW group, the sound of timber fracture could be heard after the maximum load was reached. Although the bearing capacity of these specimens decreased stepwise, their overall bearing capacity (maximum destructive force) and the

maximum spanwise midspan deflection of the specimen at the time of destruction were higher compared to the W group of specimens. This suggests that the BW specimens have improved the overall performance. The compressive strength of bamboo is higher than that of wood. Therefore, by reinforcing the compressed surface of the wood with bamboo, the compression resistance of the original surface can be improved, leading to an overall improvement in the static-bending performance of the wood.

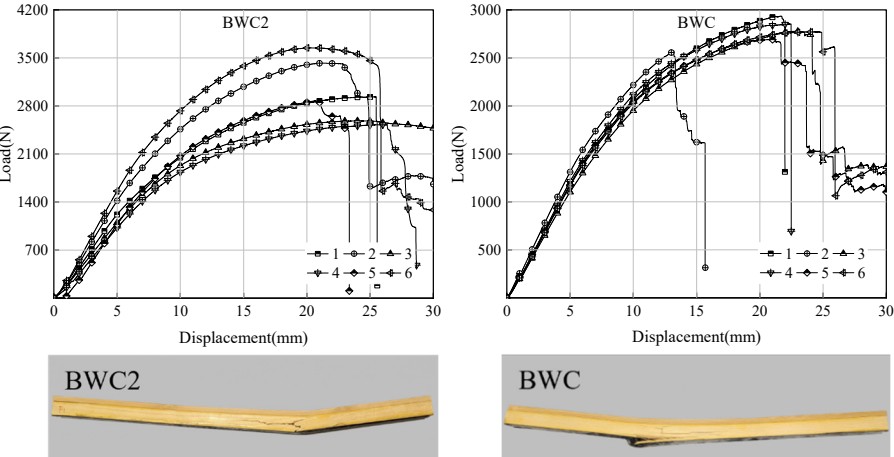

**Figure 6.** Load displacement curve and failure mode of shear failure specimen.

The damage pattern of the BWB reinforced with bamboo on both the upper and lower surfaces of the SPF timber is similar to that of two other composite materials, namely W and BW. During the loading process, the deformation between bamboo and timber is well-coordinated, with no apparent slippage. However, the maximum load required to cause damage is significantly higher in the case of BWB than W and BW. Both the parallel tensile strength and the parallel compressive strength of bamboo are higher than that of SPF, making the maximum breaking load of BWB larger. Although bamboo's parallel tensile strength is almost twice its parallel compressive strength and the parallel tensile modulus of elasticity is about seven times greater than its parallel compressive modulus of elasticity [23], the heat treatment process reduced the parallel tensile properties of bamboo, leading to brittle fracture in the BWB group.

As can be seen from Figure 5, the static-bending damage of the specimens in groups WB, WC, and WC2 can be classified as plastic fracture damage. The basic characteristic is that the "load-displacement" curve of the specimens can be clearly divided into two parts: the straight-line section (a–b section) and the curved section (b–c section). The force in sections a–b increased linearly and the specimen bending was not obvious; in sections b–c, an inelastic plateau and the specimen bending increased rapidly. In the load-displacement curves of the WB group, the force value did not decrease significantly, which indicated that the specimen fracture was mainly caused by the fracture of bamboo fibers; the force value of group WC and WC2 began to decrease slowly after reaching the peak, which indicated that fiber fracture damage made the specimen's overall stiffness decrease gradually, deformation increased gradually, the specimen ruptured until it was broken, and the maximum midspan deflection at the time of damage was more than 20 mm. All three sets of specimens exhibited obvious compression folds in the compression zone (upper surface) during damage. There are two primary reasons for the damage phenomenon mentioned above. First, the longitudinal (L) tensile strength of coniferous wood is slightly greater than the longitudinal compression strength, its tensile damage is mostly brittle fracture damage, and its compression damage is mostly folded compression plastic damage (in fact, this is the wood of the local instability of the damage rather than fracture damage). The former deformation is much smaller than the latter when the damage occurs. Second, compared with the reinforced tensile surface of the specimen, the compression surface naturally becomes the weakest region of the whole beam, and the destruction of this region

must show the typical fold-like deformation and a large amount of plastic deformation in two aspects.

Figure 6 illustrates that the primary failure modes of the specimens in the BWC and BWC2 groups were substrate shear damage, including longitudinal fiber fracture and transverse shear fracture [22]. BWC group specimens in the static-bending loading process first appeared along the length of the substrate in the horizontal shear crack, followed by the expansion of the SPF wood cracks, and longitudinal fracture of fibers on the underside of the SPF wood. But, due to the existence of a carbon-fiber-reinforced layer, it still had a certain load-bearing capacity until the SPF wood bottom fracture site pierced the carbon-fiber-reinforced layer and ultimately led to the destruction of the specimen as a whole. However, the BWC2 group specimens were strengthened by two carbon-fiber layers, which made the SPF timber bottom-surface fracture without piercing the carbon-fiber reinforcement layer. This made the whole beam still have a good load-bearing capacity until the bottom surface of the SPF timber and the carbon-fiber layer of the gluing surface of the shear were damaged and ultimately led to the destruction of the specimen as a whole. The maximum static flexural damage loads for specimens in groups BWC and BWC2 were significantly higher than for all other specimens.

The reason for this damaging phenomenon is that the compression region of these two composite materials in the process of static-bending force is borne by the bamboo material, which has a much larger compressive strength and compressive modulus of elasticity than SPF, while the tensile region is borne by the carbon fiber, which has a much larger tensile strength and tensile modulus of elasticity than SPF. And, the substrate, which has a weaker shear strength than SPF, bears the horizontal shear force of the whole beam.

It can be seen that bonding bamboo with high compression strength on the compression side and carbon fiber with a high tensile strength on the tensile side of the wood can significantly improve the static-bending properties of the wood.

### 3.2. Effect of Reinforcement Method on MOR and MOE of Specimens

Table 3 and Figure 7 provide the MOR and MOE values for each type of composite specimen created by reinforcing bamboo with CFRP. The MOR of all composite specimens was significantly enhanced after the reinforcement treatment, with a range of improvement from 15.6% to 97.2%. However, the MOE values showed mixed results, with the MOE of WB, BW, and BWB decreasing by 7.8% to 16.4%. On the other hand, the MOE of WC, WC2, BWC, and BWC2 increased by 23.7% to 49.7%.

According to the theory of the static-bending of beams, it is evident that the enhancement of the MOR of all types of composite materials is due to the parallel compressive and tensile strengths of bamboo. In addition, the parallel tensile strengths of carbon fibers are better than those of the base-material SPF. The enhancement becomes more apparent when bamboo adheres to the compression side of SPF and carbon fibers are adhered to the tensile side of SPF or a combination of both.

**Table 3.** Test results of specimens.

| Type | Original Wood MOE (MPa) | Composite Specimens MOE (MPa) | | Composite Specimens MOR (MPa) | |
|---|---|---|---|---|---|
| | Average (CoV) | Average (CoV) | Growth Rate | Average (CoV) | Growth Rate |
| W | 11,337 (0.028) | - | - | 68.42 (0.060) | - |
| WB | 12,407 (0.068) | 11,078 (0.062) | −10.7% | 79.06 (0.051) | 15.6% |
| BW | 11,854 (0.069) | 10,929 (0.085) | −7.8% | 81.26 (0.053) | 18.8% |
| BWB | 12,228 (0.139) | 10,218 (0.158) | −16.4% | 90.33 (0.129) | 32.0% |
| WC | 14,144 (0.034) | 17,489 (0.038) | 23.6% | 99.76 (0.033) | 45.8% |
| WC2 | 12,358 (0.153) | 18,499 (0.073) | 49.7% | 102.99 (0.067) | 50.5% |
| BWC | 12,440 (0.065) | 15,723 (0.050) | 26.4% | 124.51 (0.045) | 82.0% |
| BWC2 | 11,848 (0.069) | 16,974 (0.155) | 43.3% | 134.91 (0.150) | 97.2% |

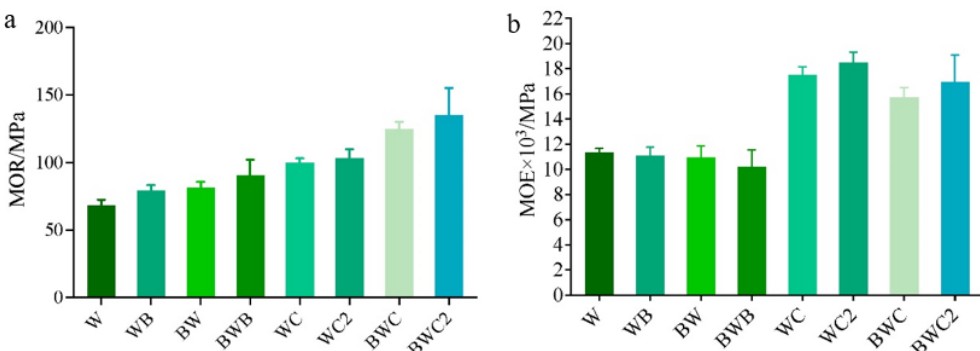

**Figure 7.** MOR (**a**) and MOE (**b**) of specimens after surface composite in different ways.

Although the MORs of WB, BW, and BWB were increased after the bamboo reinforcement treatment, the MOEs of the specimens were significantly reduced. The main reason is that, to prevent mold and insect damage to bamboo, the moso bamboo used in this experiment was heat-treated beforehand, and its MOE is only 6084 MPa, which is smaller than that of the substrate SPF MOE of 11,336 MPa. However, the SPF tensile surface reinforced by carbon fibers with a tensile modulus of up to 210,000 MPa resulted in a larger natural increase in MOE for the whole beams of WC, WC2, BWC, and BWC2. When carbon fibers with a greater parallel tensile strength and modulus of elasticity than SPF were applied to the tensile side (lower surface) of SPF, and bamboo with greater parallel compressive strength and modulus of elasticity than SPF was applied to the compressive side (upper surface), the MORs and MOEs of the whole beams of BWC and BWC2 were significantly improved compared to the control SPF. The improvement in MOR and MOE for BWC was 82.0% and 26.4%, respectively. Similarly, for BWC2, the improvement in MOR and MOE was 97.2% and 43.3%, respectively.

Overall, the asymmetric composite approach of the BWC2 specimen showed the best MOR and MOE enhancement of the substrate SPF.

### 3.3. Cross-Sectional Strain Analysis

Figure 8 shows the strain field in the L direction in the pure bending section (between the two loading points) of a representative specimen of each type that reaches the ultimate load during static bending but has not yet been damaged (critical damage point).

For both the W and BWB specimens, the maximum tensile strain in the tensile region of the control W specimens with relatively uniform material properties and the BWB specimens with top and bottom symmetric structures are slightly smaller than the maximum compressive strain in the compressive region at the time of critical damage, and the neutral axes of these two types of specimens (the black lines in the figures) are almost coincident with their horizontal geometrical central axes, as shown in Figure 8a,d. This is because the strain response in the tensile and compressive regions of both SPF, which is relatively homogeneous, and BWB, which is a composite with an upper and lower symmetric structure, is relatively homogeneous because homogeneous materials have similar elastic properties even though their tensile and compressive modulus of elasticity and tensile and compressive strengths are different. As a result, the strain distributions in the tension and compression regions are symmetrical in the static-bending state. In addition, from the comparison of Figure 8a,d, it can be seen that the maximum tensile and compressive strain before the damage of the BWB are enhanced tremendously after the upper and lower surfaces of the SPF are reinforced by the bamboo material, which is macroscopically manifested as a large increase in the MOR, as shown in Figure 7a. In Figure 8d, a large number of strain "flame-like protrusions" on the interface of bamboo and substrate SPF gluing indicate that SPF has withstood a large strain, indicating that there is a good deformation synergy between them both, reflecting a better bond between them both, which can also be seen from the specimen in Figure 4, where, after the destruction of the bamboo–wood specimens, the breakage rate is close to 100%.

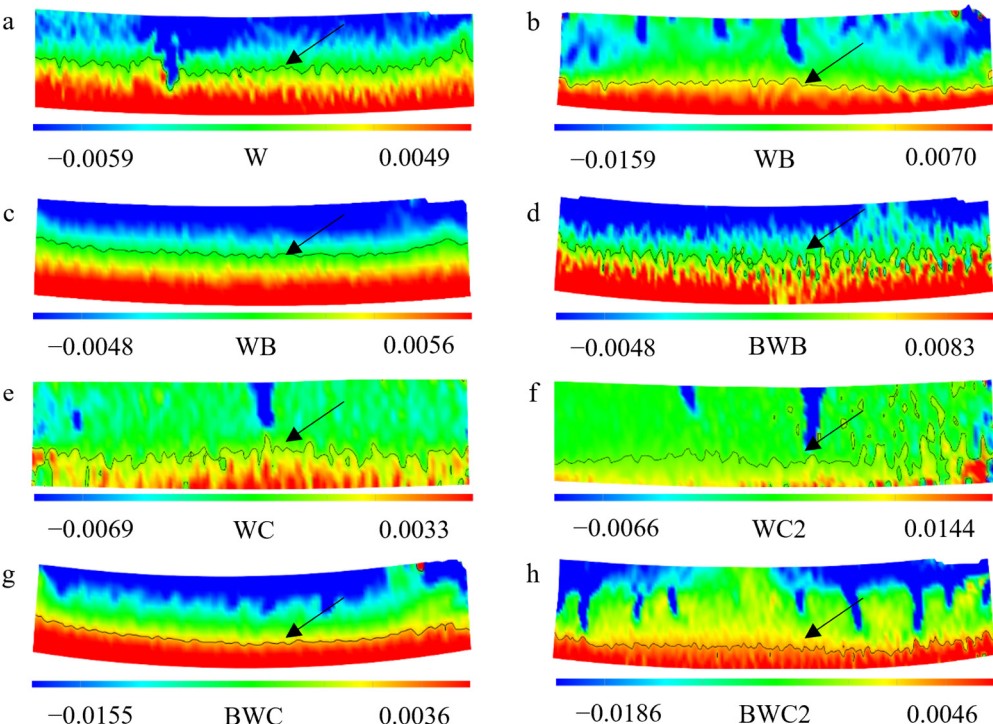

**Figure 8.** Strain in the L direction of the cross section when the force reaches the ultimate load during static bending of each type of specimen. (Note: The black arrow points to the line as the location of the cross-section neutral layer. The letters (**a**–**h**) denote the title of each figure).

For the BW specimen, its maximum compressive strain is less than its maximum tensile strain because the upper surface of the base-material SPF receives reinforcement from the bamboo material, which has a larger compressive strength and compressive modulus of elasticity in the cis-grain than that of SPF. Coupled with the fact that bamboo material is much more homogeneous than wood, its strain distribution in the compression region while obtaining MOR enhancement is much more homogeneous than the distribution of strain in the compression region of the W specimen, as shown in Figure 8c.

For the WB, WC, and WC2 specimens, the difference between the maximum compressive strain and the maximum tensile strain is much larger than that of the W specimen because the lower surface of the base-material SPF is reinforced by bamboo or CFRP, which obtains a greater compliant tensile strength than that of SPF. This also resulted in a significant downward shift of the neutral axis for these three types of specimens, as shown in Figure 8b,e,f. However, it should be noted that while these three types of specimens have achieved significant improvements in MOR and MOE, there are also very prominent compression-stress concentrations in their compression regions.

For the BWC and BWC2 specimens, due to the reinforcement of bamboo with higher compressive strength and modulus in the upper surface of the base-material SPF and CFRP with higher tensile strength and modulus in the lower surface of the base-material SPF, the difference between the maximum compressive strain and the maximum tensile strain of the two types of specimens is further enlarged compared to that of the other reinforcement modes of composite materials, which consequently leads to the production of the neutral axis of these three specimens to produce a further downward shift, as shown in Figure 8g,h. However, due to the uniformity of the reinforcing phases of bamboo and CFRP, the BWC and BWC2 specimens did not show significant compressive stress concentrations in the compression region while obtaining large MOR and MOE increases.

From the above analysis, it can be seen that the simultaneous reinforcement of the upper and lower surfaces (tensile and compressive regions) of the substrate SPF with bamboo and CFRP with strong compressive and tensile properties, respectively, on the one

hand, led to the asymmetric surface-reinforced composites BWC and BWC2 obtaining a huge MOR and MOE enhancement on the other hand; on the other hand, the stress transfer between the reinforcing material and the substrate is good, which greatly overcomes the risk of sudden brittle fracture of the substrate SPF due to the stress concentration brought about by the nonuniformity of the material properties, as shown in Figures 4 and 5, and greatly improves the safety performance of the material.

## 4. Numerical Simulation and Validation of AFRWC Static-Bending Performance

From the previous analysis, it can be seen that the AFRWC such as BWC2 is more effective in enhancing the static-bending strength and static-bending modulus of elasticity of SPF wood. In this paper, an implicit numerical model of the static-buckling behavior of BWC2-type members is developed by modeling and analyzing them using ABAQUS/CAE 2021. The arrangement of wood fibers and bamboo fibers has obvious directionality, with better mechanical properties along the fiber direction and weaker properties when transverse to the direction of the fibers; CFRP is similar to wood, with higher strength and stiffness in the direction of the fibers, and weaker properties in the transverse direction. So, SPF wood, bamboo, and CFRP are considered orthotropic materials, where the material direction of the wood is shown in Figure 3c, respectively. During the static-bending experiments, there was no significant slip observed at the glued interface. The connection that exists between timber, CFRP, and bamboo is known as a tie connection. Normal contact between the specimen and the loading point is hard contact; tangential contact is frictionless. Limit the displacement in the Z direction at both ends of the specimen and the direction of rotation around the X and Y axes when setting the boundary constraints. Disregarding the effect of defects in the wood and bamboo itself, C3D8R eight-node linear hexahedral cells were selected for meshing the wood and bamboo, while S4R four-node curved thin-shell cells were selected for meshing CFRP. By comparing the effect of different sizes of the cell on the calculation results, it is concluded that when the cell size was limited below 3 mm × 3 mm × 3 mm, both the calculation accuracy and the calculation time can be accepted.

Each material requires a total of nine independent material elastic constants for the modulus of elasticity, Poisson's ratio, and shear modulus in the longitudinal (L), radial (R), and tangential (T) directions, respectively, based on the elasticity parameters of the wood for SPF, as given in the literature [17], and the elasticity constants of bamboo and CFRP, as given in the literature [24,25] for the three materials that are shown in Table 4.

**Table 4.** Elastic constants of materials.

| Type | $E_L$ /MPa | $E_R$ /MPa | $E_T$ /MPa | $\mu_{LT}$ | $\mu_{LR}$ | $\mu_{TR}$ | $G_{LT}$ /MPa | $G_{LR}$ /MPa | $G_{RT}$ /MPa |
|------|------------|------------|------------|------------|------------|------------|---------------|---------------|---------------|
| SPF | 11,336 | 998 | 499 | 0.347 | 0.315 | 0.408 | 918 | 1088 | 125 |
| Bamboo | 6084 | 1710 | 1177 | 0.273 | 0.291 | 0.268 | 575 | 549 | 325 |
| CFRP | 105,000 | 7200 | 7200 | 0.340 | 0.340 | 0.378 | 3400 | 3400 | 2520 |

Note: E is the modulus of elasticity; G is the shear modulus; $\mu_{LT}$—*L* denotes the direction of applied stress and *T* denotes the direction of lateral deformation.

Figure 9 shows the comparison between the "load-displacement" curves of the BWC2 specimen in static bending and the measured curves given by the simulation of ABAQUS finite element software. Among them, the simulated values of MOR and MOE for the BWC2 specimen were 133 MPa and 15,038 MPa, respectively, while the errors between the experimentally measured results and the simulated values were only −0.7% and −7.3%, respectively. The MOE simulation results of BWC2 are slightly biased due to the fact that the bamboo material is affected by the bamboo joints and the location where the material is taken, leading to a large variation in its MOE.

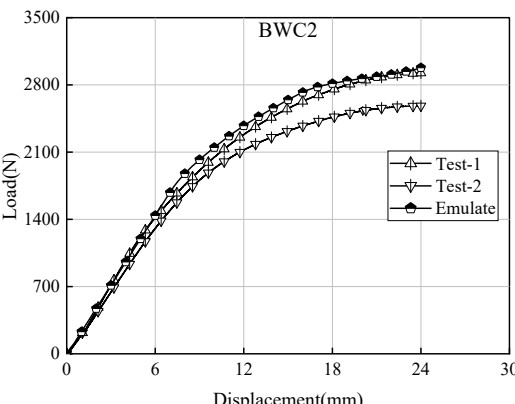

**Figure 9.** Comparison between simulation and experimental results of load-displacement curves during static bending of BWC2.

Figure 10 shows the strain-field cloud of BWC2 in the L direction under ultimate load during static bending simulated by the simulation. From Figure 10, it can be seen that the bamboo layer carries the maximum compressive stress/strain of the whole beam while the carbon-fiber layer carries the maximum tensile stress/strain of the whole beam. Comparing Figure 8a and the strain-field maps of the three materials in the pure bending section, it can be seen that, even under the larger static-bending load (2979 N), the maximum compressive and tensile strain on the upper and lower surfaces of the SPF as the substrate are much smaller than those on the upper and lower surfaces of the SPF timber when it is loaded by itself (with the maximum load of 1633 N). The eventual damage of the BWC2 becomes shear damage of SPF with the substrate type of shear damage. The final damage of BWC2 became SPF parallel grain shear damage, and the damage type was matrix shear damage. By using asymmetric surface reinforcement of bamboo and carbon-fiber materials, the tensile and compressive strains on the surface of the wood can be effectively reduced. This results in a significant improvement in both static-bending properties and the overall safety of wood and can provide a new path for the industry to develop high-performance and high-safety load-bearing wood-based composites.

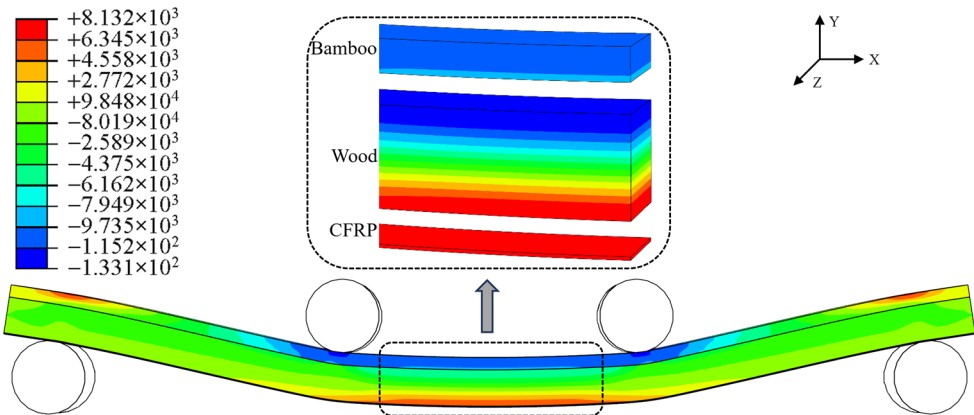

**Figure 10.** Strain-field nephogram of BWC2 under ultimate loading.

## 5. Conclusions and Discussion

The study aimed to investigate how enhancing the static-bending properties of wood can be achieved by using asymmetric surface reinforcement of raw bamboo and carbon-fiber materials. The researchers conducted four-point bending experiments and DIC analyses to compare the results with wood that had been reinforced with four materials, as well as wood that had been reinforced with a single material on its surface. Based on the findings, the researchers drew several conclusions.

(1)     The types of static flexural damage of each type of AFRWC specimens and control W specimens formed by bamboo and CFRP reinforcement are three types: brittle fracture damage, plastic fracture damage, and matrix shear damage;

(2)     The MOR of each type of AFRWC specimen ranged from 15.6% to 97.2% higher than the control W, while the MOE increased and decreased, with the MOEs of WB, BW, and BWB decreasing by 7.8%–16.4%, but the MOEs of WC, WC2, BWC, and BWC2 increasing by 23.7%–49.7%;

(3)     In the BW, BWC, and BWC2 type specimens, bamboo and carbon-fiber materials effectively reduced the compressive and tensile strains of wood during static bending, respectively, and reduced the compression damage caused by stress concentration. There is an obvious downward shift of the neutral axis in the static-bending process of the WB, WC, WC2, BWC, and BWC2 specimens.

(4)     BWC and BWC2, which are asymmetrically reinforced AFRWC specimens, obtained significant MOR and MOE enhancement. They also overcome the risk of sudden brittle fracture of the base-material SPF due to stress concentration caused by uneven material properties, which greatly improves the safety performance of the material;

(5)     Based on the simulation of ABAQUS finite element software, the "load-displacement" curves of the BWC2 specimen in the process of static-bending force are in good agreement with the measured curves, and the simulated values of MOR and MOE of the BWC2 specimen are 133 MPa and 15,038 MPa, respectively, with the errors of only −0.7% and −7.3%, respectively, compared to the experimental results. The MOR and MOE were 133 MPa and 15,038 MPa, respectively, and the errors between them and the measured results were only −0.7% and −7.3%, respectively.

There are two crucial matters that require consideration in this paper. The dimensions of the specimens used in this study are only 400 mm (L) × 20 mm (T) × 20 mm (R), and there are two reasons for such a small size. This is mainly based on two considerations; one is that full-size specimens will inevitably be affected by defects, such as knots, cracks, and other defects, in the examination of the reinforcing materials and the effects of various types of asymmetric reinforcement on the full-field strain of the wood stress cross section, which is not conducive to the study of the basic laws of the asymmetric reinforcement method. Secondly, the test results obtained from small-size specimens applied to full-size wood must be considered in the size-effect problem, and this problem is different defects and the distribution of the mechanical properties of the wood-loss effect. In this regard, there are several scholars who have done a lot of research [26–28], and there are already some more mature and reliable size adjustment coefficients that can be used to assess this effect [29,30].

Another issue is related to the numerical simulation of the static-bending performance of BWC2-type AFRWC. In this study, based on the experimental results and analysis, only the elastic and elastoplastic static-bending properties of the BWC2 specimen with the best asymmetric reinforcement were selected for numerical simulation, and its fracture behavior was not simulated. The main reason is that the fracture damage of BWC2 may be the gluing damage among bamboo, carbon-fiber reinforcement and wood, the compression damage of wood (Figure 8h), or the mixed damage with both kinds of damage, which is difficult to simulate with the help of the existing fracture-damage simulation of a single material in theory and practice. This also highlights the need for in-depth research on AFRWC fracture damage in the future.

**Author Contributions:** Conceptualization, X.G. and C.D.; methodology, X.G.; software, X.G.; validation, X.G. and F.Z.; formal analysis, X.G. and F.Z.; data curation, F.Z.; writing—original draft preparation, X.G.; writing—review and editing, C.D.; visualization, S.D.; supervision, S.D.; funding acquisition, S.D. and C.D. All authors have read and agreed to the published version of the manuscript.

**Funding:** This research was funded by the National Natural Science Foundation of China (No. 31960291), Yunnan Agricultural Joint Specialization (No. 202301BD070001-060), Yunnan Provincial Reserve Program for Young and Middle-aged Academic and Technical Leader (No. 202305AC160074), and the Yunnan Province Natural Science Key Foundation (No. 202201AS070152).

**Data Availability Statement:** The raw/processed data required to reproduce these findings cannot be shared at this time, as the data also forms part of an ongoing study.

**Conflicts of Interest:** The authors declare no conflict of interest.

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
