# Peer review of "Study on the Static-Bending Properties of Surface-Reinforced Wood with Asymmetric Fibers"

_forests, doi:10.3390/f14122454_

Round 1

Reviewer 1 Report

Comments and Suggestions for Authors

Review: Study on the Static Bending Properties of Surface-reinforced Wood with Asymmetric Fibers

Journal: Forests (ISSN 1999-4907)

Manuscript ID: forests- 2752901

1.     Abstract: The word count in abstract exceeds the maximum limit of 200 words, as described in the guidelines.

2.     Introduction / at the end: It is necessary to more clearly define the problem that the research aims to solve and clearly state the research question. Additionally, presenting how your research builds upon existing knowledge and the trajectory of research in the field can help readers better understand the contribution of your work.

3.     Discussion is missing from the article and it is required by Instructions for Authors. Authors should discuss the results and how they can be interpreted from the perspective of previous studies and of the working hypotheses. The findings and their implications should be discussed in the broadest context possible.

The study mentions significant improvements in MOR for AFRWC but does not explicitly address the generalizability of these findings to different wood species or environmental conditions. Discussing the applicability and potential limitations of the results would enhance the study's practical implications.

Asymmetric fiber reinforcement involving bamboo and carbon fiber may raise concerns about the cost and practicality of widespread implementation. Including a brief discussion on the economic feasibility and practical considerations would provide a more holistic view.

4.     The conclusion needs refinement.

·       The conclusion should be improved by emphasizing practical implications and future research directions. Practical Implications: Highlight the key practical benefits arising from the improvement of the static properties of wood based on your work. Discuss the possibilities of applying reinforced materials in real construction projects and the wood industry.

·       Identify possible challenges or limitations that this research has uncovered and propose ways to overcome them. Specify the necessary further steps for a deeper understanding of the specific reinforcement mechanisms and to achieve even better results. Suggest possible variations in reinforcement methods or new material combinations for exploration. Express the overall importance of this research in the context of developing sustainable and high-performance materials in the construction industry.

Author Response

I have responded to all of your comments. Please refer to the attachment for more information. Thank you.

Reviewer 2 Report

Comments and Suggestions for Authors

The article contains very important errors that are worth clarifying. At the beginning, small-sized wooden elements cannot be compared to full-size structural elements, even due to the scale effect, because it is a heterogeneous element. Then there is a lack of research methodology, as well as preparation, what the tensioning force looked like and how much it was, etc. And what did the tensioning look like? Then there is no information about the next stages of work. The numerical results can also be compared with the experimental results.

Comments on the Quality of English Language

Moderate editing of English language required

Author Response

(The authors gave the same response as above.)

Reviewer 3 Report

Comments and Suggestions for Authors

The paper deals with the static bending behavior of fiber reinforced wood composite. However, they have just investigated a kind of sandwich panels made of simple wooden core with bamboo skin. Fiber reinforced wood composite is somehow different from the author’s proposed panel. The above assumption can be considered as major flaw in the article. Based on the criteria, the article in the present form is not suitable for publication and the article must be deeply modified with the following comments for the consideration of publication.

Comments on the Quality of English Language

Moderate editing of English language required

Author Response

(The authors gave the same response as above.)

Round 2

Reviewer 2 Report

Comments and Suggestions for Authors

No comments.

Author Response

I have made some revisions according to the comments of another reviewer, please check it, thank you.

Reviewer 3 Report

Comments and Suggestions for Authors

Author needs to deeply modify  the numerical simulation part of the article. Please find the comments attached.

Comments on the Quality of English Language

Usage of english language is in good form.

Author Response

I have made changes to the article in response to the questions and comments you gave and made a relevant response, please check, thank you!
